

# Benchmarking protocols for the metagenomic analysis of stream biofilm viromes

Meriem Bekliz, Jade Brandani, Massimo Bourquin, Tom J. Battin and Hannes Peter

Stream Biofilm and Ecosystem Research Laboratory, École Polytechnique Federale de Lausanne, Lausanne, Switzerland

## ABSTRACT

Viruses drive microbial diversity, function and evolution and influence important biogeochemical cycles in aquatic ecosystems. Despite their relevance, we currently lack an understanding of their potential impacts on stream biofilm structure and function. This is surprising given the critical role of biofilms for stream ecosystem processes. Currently, the study of viruses in stream biofilms is hindered by the lack of an optimized protocol for their extraction, concentration and purification. Here, we evaluate a range of methods to separate viral particles from stream biofilms, and to concentrate and purify them prior to DNA extraction and metagenome sequencing. Based on epifluorescence microscopy counts of viral-like particles (VLP) and DNA yields, we optimize a protocol including treatment with tetrasodium pyrophosphate and ultra-sonication to disintegrate biofilms, tangential-flow filtration to extract and concentrate VLP, followed by ultracentrifugation in a sucrose density gradient to isolate VLP from the biofilm slurry. Viromes derived from biofilms sampled from three different streams were dominated by *Siphoviridae*, *Myoviridae* and *Podoviridae* and provide first insights into the viral diversity of stream biofilms. Our protocol optimization provides an important step towards a better understanding of the ecological role of viruses in stream biofilms.

## INTRODUCTION

Viruses are the smallest and most abundant biological entities on Earth, typically outnumbering their prokaryotic and eukaryotic hosts by an order of magnitude (*Rohwer, 2003*; *Sime-Ngando, 2014*). Viruses are a large reservoir of genetic diversity (*Suttle, 2007*; *Sullivan, Weitz & Wilhelm, 2017*; *Angly et al., 2006*; *Brum et al., 2015*) and occur in all habitats (*Paez-Espino et al., 2016*), including air (*Reche et al., 2018*; *Rosario et al., 2018*), soils (*Srinivasiah et al., 2008*; *Williamson et al., 2017*) and deep-sea sediments (*Danovaro et al., 2008*). The study of viral ecology was pioneered in surface marine systems where viruses that infect bacteria, also known as bacteriophages, are the main source of bacterial mortality (*Suttle, 2007*; *Brum & Sullivan, 2015*; *Gregory et al., 2019*) and impact ecosystem functions such as the cycling of carbon (*Breitbart et al., 2018*). By lysing their hosts,

Corresponding author
Hannes Peter, hannes.peter@epfl.ch

horizontal gene transfer and metabolic reprograming, bacteriophages play a pivotal role in structuring microbial communities (*Skvortsov et al., 2016*; *Silva et al., 2017*; *Rossum et al., 2018*; *Daly et al., 2019*), the flow of energy and matter through food webs (*Weitz et al., 2015*), the cycling of carbon and nutrients (*Dell'Anno, Corinaldesi & Danovaro, 2015*; *Guidi et al., 2016*; *Emerson et al., 2018*) and the evolution of bacteria (*Pal et al., 2007*; *Rodriguez-Valera et al., 2009*; *Simmons et al., 2019*). Many of these strides have only recently been possible with the advent of molecular tools such as metagenomic sequencing (*Rosario & Breitbart, 2011*; *Brum & Sullivan, 2015*; *Roux et al., 2016*, *2017*). In this context, one may differentiate between untargeted approaches, which have produced a wealth of viral sequences (*Paez-Espino et al., 2016*) and virome sequencing, in which the viral fraction is purified prior to sequencing, thus ensuring that the sequencing effort is targeted towards viral nucleic acids (*Rosario & Breitbart, 2011*).

Besides the early recognition of the potential of phages to eradicate bacterial biofilms in medical settings (*Chan & Abedon, 2015*), little is known about the interactions between biofilms and viruses (*Sutherland et al., 2004*). While the biofilm matrix may impede the access of viruses to the surface of bacterial cells (*Vidakovic et al., 2017*), the susceptibility to phage-induced clearance of biofilm has been demonstrated in laboratory experiments (*Scanlan & Buckling, 2012*). Biofilms may also act as a reservoir for virus amplification and viruses may endure periods of unfavorable environmental conditions within the extracellular matrix (*Doolittle, Cooney & Caldwell, 1996*; *Briandet et al., 2008*). Apart from a few examples (*Dann et al., 2016*; *Silva et al., 2017*; *Rossum et al., 2018*), we lack an understanding of the ecological role of viruses in streams and rivers in general (*Peduzzi, 2016*) and in stream biofilms in particular (*Battin et al., 2016*). In streams, biofilms colonize the sedimentary surfaces of the streambed, are biodiversity hotspots comprising members from all three domains of life and fulfill critical ecosystem processes (*Battin et al., 2016*). It is reasonable therefore to speculate that viruses also control biodiversity and biomass turnover in stream biofilms with potential consequences for ecosystem functioning and biogeochemical cycling. However, the heterogeneous matrix of stream biofilms has precluded the study of viromes in stream biofilms.

In general, viral metagenomics is complicated by the vast diversity of viruses and lack of universal marker genes, the risk of contamination with non-viral DNA and the underrepresentation of viral sequences in databases (*Thurber et al., 2009*; *Hayes et al., 2017*). Protocols for sample preparation for viral metagenomics therefore aim at concentrating and purifying viral-like particles (VLPs) and removing contaminating DNA, while optimizing VLP recovery (*Castro-Mejia et al., 2015*). However, it is clear that viruses are lost at every step of these protocols. Large viruses, in the size range typical for bacteria, may be removed by filtration. Some viruses are sensitive to chemicals during purification based on the structure of their capsids, while other viruses may be lost because of differences in buoyant density, critical during purification in density gradient ultracentrifugation (*Thurber et al., 2009*). It is thus clear that no single protocol to extract all viruses exists and modifying as well as rigorous testing of protocols remains a critical task.

**Table 1 Sample site and biofilm characteristics.**

|  | VDN | VEV | SNG |
|---|---|---|---|
| Coordinates | 46°15′13.5″N 7°06′33.9″E | 46°30′46.4″N 6°54′43.7″E | 46°33′23.9″N 6°28′55.3″E |
| Altitude (m a.s.l.) | 1,210 | 766 | 498 |
| Bacterial abundance (cells m$^{-2}$) | $2.3 \times 10^9$ | $1.8 \times 10^{10}$ | $4.1 \times 10^{11}$ |
| Chlorophyll-a (µg cm$^{-2}$) | 0.21 | 0.22 | 1.21 |
| EPS proteins (µg cm$^{-2}$) | 0.05 | 0.04 | 0.24 |
| EPS carbohydrates (µg cm$^{-2}$) | Below detection | Below detection | 0.08 |

The first critical step towards the study of biofilm viruses in streams using virome sequencing requires the effective extraction of VLPs from the biofilm matrix. The aim of our study was therefore to optimize a sample-to-sequence pipeline including VLP extraction, concentration and purification towards metagenomic analyses. Our effort was specifically tailored to maximize the yield of viral DNA while minimizing DNA contamination from bacteria, eukaryotic hosts or the environment. To this end, we compared a suite of sequential protocols to generate viral metagenomes from stream biofilms including tangential flow filtration (TFF), polyethylene glycol (PEG) precipitation, physicochemically induced biofilm breakup and ultracentrifugation followed by nucleic acid extraction. Several of these protocols have been described previously (*Danovaro et al., 2001*; *Thurber et al., 2009*; *Danovaro & Middelboe, 2010*; *Hurwitz et al., 2013*; *Temmam et al., 2015*; *Trubl et al., 2016*; *Hayes et al., 2017* for reviews), but their rigorous testing for virome generation from stream biofilms is lacking at present. We benchmark the efficiency of the different protocols using epifluorescence microscopy counts of VLP and DNA yields and provide first results of the virome structure from biofilms in three streams. We provide a step-by-step version of the optimized protocol at protocols.io, which allows for community participation and continuous protocol development.

## MATERIALS AND METHODS

### Sampling

We sampled benthic biofilms from three streams (Switzerland) draining catchments differing in altitude and land use (Table 1). The Vallon de Nant (VDN) catchment is pristine with vegetation dominated by alpine forests and meadows. The Veveyse (VEV) catchment is characterized by mixed deciduous forests, but also agricultural and urban land use. The Senoge (SNG) catchment is clearly impacted by agricultural land use. During winter, benthic biofilms were randomly collected from stones (5–15 cm in diameter) using sterile brushes and Milli-Q water. Depending on biofilm thickness, we scraped biofilms from cobbles with a total surface area ranging from 1.3 to 2.4 m$^2$ into 10 L Milli-Q water. Slurries were transported on ice to the laboratory pending further processing.

### Biofilm properties

For bacterial cell counting, samples were fixed with 3.7% formaldehyde (final concentration) and stored at 4 °C. Bacterial cells were disintegrated from the biofilm

matrix using 0.25 mM tetrasodium pyrophosphate in combination with rigorous shaking (1 h) and sonication (Bransonic Sonifier 450, Branson, MO, USA) on ice (1 min) (*Velji & Albright, 1993*). As previously established (*Besemer et al., 2009*), cells were stained using Syto13 and counted on a flow cytometer (NovoCyte, ACEA Biosciences, San Diego, CA, USA). Chlorophyll *a* content was determined spectrophotometrically after over-night extraction in 90% EtOH (*Lorenzen, 1967*). Extracellular polymeric substances (EPS) were extracted from biofilm slurries using 50 mM EDTA and shaking (1 h) (*Battin et al., 2003*). Carbohydrates were precipitated in 70% EtOH (−20 °C, 48 h) and measured spectrophotometrically as glucose equivalents (*DuBois et al., 1956*). Proteins were determined according to *Lowry et al. (1951)*.

## Transmission electron microscopy

A first confirmation of the presence of VLP in biofilms was obtained from TEM. For this, 5 μL of unprocessed sample was adsorbed to a glow-discharged carbon-coated copper grid (Canemco & Marivac, Gore, QC, Canada), washed with deionized water and stained with 5 μL of 2% uranyl acetate. TEM observations were made on an F20 electron microscope (ThermoFisher, Hillsboro, OR, USA) operated at 200 kV and equipped with a $4{,}098 \times 4{,}098$ pixel camera (CETA, ThermoFisher, Waltham, MA, USA). Magnification ranged from 10,000 to 29,000×, using a defocus range of −1.5 to −2.5 μm.

## Protocols for the extraction of VLP

In order to establish an optimized stream biofilm sample-to-sequence pipeline, we explored a variety of protocols for the concentration, extraction and purification of VLP (Fig. 1). The pipeline consists of three main parts: concentration, extraction and purification. The concentration step is required to obtain sufficient genetic material for nucleic acid extraction. Extraction aims to liberate VLP from the biofilm matrix, while purification aims to reduce the amount of contaminant nucleic acids and cellular debris. The order—first volume reduction, then chemical detachment—was chosen because chemicals used for extraction of VLP may damage tangential flow filters. We tested all possible combinations of protocols to identify the most promising laboratory pipeline for the preparation of viral metagenomes.

First, we homogenized the biofilm slurries by manual shaking and split samples into aliquots (1 L). Aliquots were centrifuged at $100{\times}g$ (15 min) (5810R; Eppendorf, Hamburg, Germany) to remove large sediment particles and organisms contained within the biofilm slurry. We recovered the supernatant for downstream analyses.

### Concentration

We evaluated TFF and PEG precipitation to concentrate the viral size fraction. A medium-scale TFF system equipped with a 100 kDa tangential flow filter (GE Healthcare, Chicago, IL, USA) was used (*Thurber et al., 2009*). Viruses were collected in the retentate, whereas water and particles smaller than the pore size were discarded. For each aliquot, the initial volume was reduced to less than 50 mL.

PEG precipitation was performed as described previously (*Bibby & Peccia, 2013*). Aliquot biofilm samples were supplemented with 10% w/v PEG 8000 (Sigma–Aldrich,

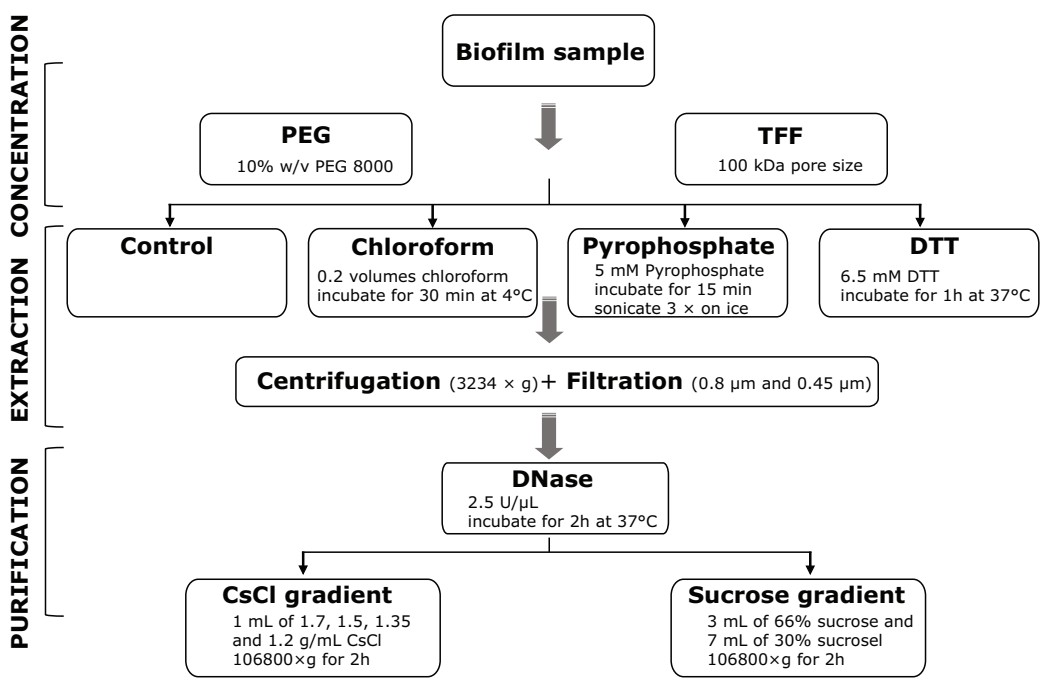

**Figure 1 Overview of methods for the extraction and purification of viruses from stream biofilms.**
First, phages are concentrated using either PEG precipitation or TFF. Different physico-chemical
extraction procedures were then evaluated for their efficiency. Prior to DNase I digestion, centrifugation
and filtration was used to remove cell debris from all samples. Finally, ultracentrifugation in sucrose or
CsCl density gradients was used to purify viruses for downstream molecular analyses. Combinations of
all protocols were evaluated for the recovery of VLPs and DNA yield.

Steinheim, Germany) and 2.5 M NaCl (Sigma–Aldrich, Steinheim, Germany). The
supernatant was then agitated by inverting the tubes three times and stored overnight
(4 °C), followed by centrifugation for 30 min at 9,432×g at 4 °C (Sorvall RC-5C centrifuge,
HS-4 rotor; ThermoFisher, Waltham, MA, USA). Supernatants were carefully removed
and the resulting pellets were eluted with 50 mL sterile $H_2O$.

### Extraction

We tested three different physicochemical treatments (chloroform, tetrasodium
pyrophosphate in combination with sonication and dithiothreitol (DTT)) to dislodge
viruses from the biofilm matrix. To one subset of the samples, we added 0.2 volumes of
chloroform and mixed by inversion. Then, these samples were incubated at 4 °C for
30 min, vortexed every 5 min and centrifuged at 3,234×g for 15 min at 4 °C (5810R;
Eppendorf, Hamburg, Germany) to recover the supernatant (*Marhaver, Edwards &
Rohwer, 2008*; *Thurber et al., 2008*).

Following the recommendation of *Danovaro & Middelboe (2010)*, we used sonication in
combination with tetrasodium phosphate to separate viruses from biofilms. For this,
we added five mM of tetrasodium pyrophosphate (final concentration) to another subset of
concentrated biofilm samples and incubated them in the dark (15 min). Then, all samples
were sonicated (frequency: 40 kHz; Bransonic Sonifier 450, Branson, MO, USA) three

times for 1 min with 30-s intervals during which the samples were manually shaken. To prevent heating, samples were kept on ice during sonication. Finally, a subset of the samples was treated with 6.5 mM DTT and incubated for 1 h at 37 °C (*Lim et al., 2014*). At the end of the incubation period, samples were chilled on ice. Biofilm samples without chemical treatment served as controls. After each treatment, these samples were first centrifuged (4 °C) at 3,234×*g* for 15 min (5810R; Eppendorf, Hamburg, Germany) and the supernatant was sequentially filtered through 0.8 μm and 0.45 μm filters (Whatman, Maidstone, UK; GE Healthcare, Chicago, IL, USA) to remove debris and cells.

### Purification

To eliminate contaminating eukaryotic, prokaryotic and extracellular nucleic acids, DNase I at a final concentration of 2.5 U/μL (Life Technologies, Darmstadt, Germany) and Tris buffer (100 mM Tris pH 7.5, five mM $CaCl_2$ and 25 mM $MgCl_2$) were added to the clarified supernatant and incubated at 37 °C (2 h). To confirm the removal of contaminant DNA, polymerase chain reaction (PCR) targeting the 16S rRNA gene was performed using the universal primer pair 341f (5′-CCTACGGGNGGCWGCAG-3′) and 785r (5′-GACTACHVGGGTATCTAAKCC-3′) (*Thijs et al., 2017*). PCR amplification was performed with a Biometra Thermal Cycler using Taq ALLin polymerase (Axon Lab, Baden, Switzerland) according to the manufacturer's instructions with an initial enzyme activation step (95 °C for 2 min) followed by 30 cycles of denaturation (95 °C for 30 s) and hybridization (72 °C for 30 s) and a final elongation (72 °C for 10 min). PCR products were visualized under UV light after migration on an agarose gel stained using GelRed (Biotium, Hayward, CA, USA). We did not assess the potential of PCR inhibition.

For final purification of the viral fraction, two protocols of density gradient ultracentrifugation, sucrose and CsCl density gradients, were assessed. The sucrose density gradient was achieved by three mL of 0.2-μm filtered 66% sucrose and seven mL of 0.2 μm filtered 30% sucrose (*Temmam et al., 2015*). A subset of sample was deposited on top of the sucrose density gradient and centrifuged at 106,800×*g* (Optima XPN-80 Ultracentrifuge, 32 SWi; Beckman Coulter, Indianapolis, IN, USA) for 2 h at 4 °C. The viral fraction was harvested by retrieving 1.5 mL from the interface between both layers using an 18G needle. Similarly, subsamples were deposited onto CsCl density gradients composed of layers of one mL of 1.7, 1.5, 1.35 and 1.2 g/mL CsCl. The purified viral fraction was harvested from just below the interface between the 1.5 and 1.7 g/mL CsCl layers by retrieving 1.5 mL using an 18G needle.

## Enumerating VLPs using epifluorescence microscopy

To assess the relative viral extraction efficiency of the various protocols, we counted VLP under an epifluorescence microscopy (AxioImager Z2, Zeiss, Germany) as described by *Patel et al. (2007)*. Because of high background noise owing to the biofilm matrix constituents, samples could only be counted after the purification steps. It should be noted that pyrophosphate concentrations >10 mM tend to interfere with VLP counting (*Danovaro et al., 2001*); however, we used five mM pyrophosphate during VLP liberation from biofilms. Subsamples were fixed with formaldehyde, stained with 0.5 μL of

1,000 × SYBR Gold (Molecular Probes; ThermoScientific, Waltham, MA, USA) and incubated at room temperature in the dark (30 min). After incubation, each sample was filtered onto a 0.02 μm pore size membrane filter (Anodisc; Whatman, Maidstone, UK). The filters were mounted on glass slides with a drop of VECTASHIELD antifade mounting medium (Vector Laboratories, Burlingame, CA, USA). VLP were visualized using blue light (488 nm) excitation and green (512 nm) emission. For each sample, 15–20 randomly selected images were acquired with a camera (Axiocam 506 mono, Zeiss, Germany) mounted onto the microscope. VLPs were discriminated from bacteria by size (0.015–0.2 μm) and enumerated using a custom script in Fiji (*Schindelin et al., 2012*).

## Nucleic acid extraction

To further assess the efficiency of the tested protocols to obtain viral nucleic acids for metagenome sequencing, we extracted DNA and quantified its concentration. Following *Sambrook, Fritsch & Maniatis (1989)*, 0.1 volume of TE buffer, 0.01 volume of 0.5 M EDTA (pH 8) and 1 volume of formamide was added to purified viral samples, and incubated at room temperature (30 min). Then, two volumes of cold 100% EtOH were added and incubated for 30 min (4 °C). Samples were centrifuged at 17,000×$g$ (4 °C, 20 min) and pellets washed twice with 70% cold EtOH. Pellets were air-dried and resuspended in 567 μL of TE buffer (10 mM Tris and one mM EDTA (pH 8.0)). Thirty μL of warm 10% SDS and three μL of proteinase K (20 mg/mL) were added and samples were incubated for 1 h at 37 °C. Next, 100 μL of five M NaCl and 80 μL of warm hexadecyltrimethylammonium bromide (CTAB) were added and incubated for 10 min (65 °C) (*Doyle & Doyle, 1987*). DNA was extracted in a series of chloroform, phenol: chloroform:isoamyl alcohol (25:24:1) and chloroform treatments, with centrifugation at 16,000×$g$ (10 min) for phase separation. Finally, DNA was precipitated overnight in isopropanol (−20 °C). DNA was concentrated by centrifugation at 16,000×$g$ (4°C) for 20 min, and the pellet washed twice with cold 70% EtOH, air-dried, resuspended in 50 μL nuclease-free H$_2$O and stored (−20 °C). DNA concentration was measured using Qubit and the dsDNA high-sensitivity kit according to the manufacturer's instructions (Life Technologies, Carlsbad, CA, USA).

## Library construction and sequencing

Based on the epifluorescence microscopy counts of VLP and the DNA concentration from purified biofilm samples, we selected samples processed with the best performing pipeline for metagenome sequencing. For sequencing library construction, DNA was sheared with an S2 focused ultrasonicator (Covaris, Woburn, MA, USA) to achieve a target size of DNA fragments of around 350 bp. We opted for the ACCEL-NGS® 1S PLUS DNA library kit (Swift Biosciences, Ann Arbor, MI, USA) which allows low quantities of both single- and double-stranded DNA as input (*Roux et al., 2016*). Library construction and multiplexing was performed following the manufacturer's instructions for DNA inputs (<1 ng/μL) and 20 cycles of indexing PCR. Paired-end sequencing (2 × 300 bp) was performed on a MiSeq System (Illumina, San Diego, CA, USA) at the Lausanne Genomic

Technologies Facilities. Raw sequences have been submitted to the European Nucleotide Archive under accession number PRJEB33548.

## Bioinformatic analyses

The number of reads and GC content of each sample before and after quality control were calculated using a custom python script. For virome classification, we followed the recommendations to assemble viral contiguous sequences (contigs) according to *Roux et al. (2019)*. First, BBDuk (v35.79) was used to remove Illumina adapters, filtering and trimming (trimq = 12). Next, reads with >93% similarity to a human reference genome were discarded (using BBmap). The ACCEL-NGS® 1S Plus library preparation kit includes a low complexity adaptase tail, which was clipped (10 bases) according to the manufacturer's instructions. We then used the error correction capability of Tadpole (v. 37.76) to correct for sequencing errors (mode = correct ecc = t prefilter = 2). Prior to assembly, we de-duplicated our datasets using clumpify (v37.76) with parameters set such that identical reads were identified and only one copy was retained (dedupe subs = 0). Finally, we used the SPAdes assembler (v3.13.0, *Bankevich et al., 2012*) in single-cell (—sc) mode, with error correction disabled (—only-assembler) and kmers set to 21, 33, 55, 77, 99, 127 to assemble contigs. We co-assembled the paired-end reads from both sequencing runs for each sample individually. To obtain an overview of the potential contaminant sequences (e.g., human, bacterial and phiX), we uploaded the quality-trimmed and de-replicated reads (forward orientation only) to the web interface of taxonomer (*Flygare et al., 2016*). Taxonomer is an ultrafast taxonomy assigner, which assigns and classifies reads to human, bacterial, viral, phage, fungal, phix, ambiguous (i.e., reads fit to more than one bin) and "unknown" bins. For identification and classification of viral contigs, we used MetaPhinder (v2.1, *Jurtz et al., 2016*) through the web interface hosted at the Center for Genomic Epidemiology at the Danish Technical University (DTU). We mapped the reads to the contigs using BWA-MEM (http://bio-bwa.sourceforge.net/) using default setting (e.g., a mismatch penalty of four and a minimal alignment score of 30) and then counted the number of reads mapping each contig using samtools (*Li et al., 2009*). The average number of reads per contig were adjusted by contig length and normalized by library size to obtain the average percent of reads mapping to contigs. In order to specifically target ssDNA contigs (*Trubl et al., 2019*), we matched the contigs to the Viral_rep and Phage_F domains of the PFAM database using hmmsearch (HMMER v3, *Eddy, 2011*) with cutoff scores ≥50 and $e$-values ≤ 0.001.

# RESULTS

## Biofilm properties

Bacterial abundance ranged between $4.1 \times 10^{11}$ cells m$^{-2}$ at the lowest stream (SNG) and $2.3 \times 10^{9}$ cells m$^{-2}$ at the uppermost stream (VDN; Table 1). This pattern was mirrored by an increase in chlorophyl $a$ content and the protein and carbohydrate concentrations in EPS that were higher in the lowland than the high-altitude stream (Table 1). Despite these differences in biofilm properties, no differences in extraction efficiency among the different protocols were detected. In fact, the average number of

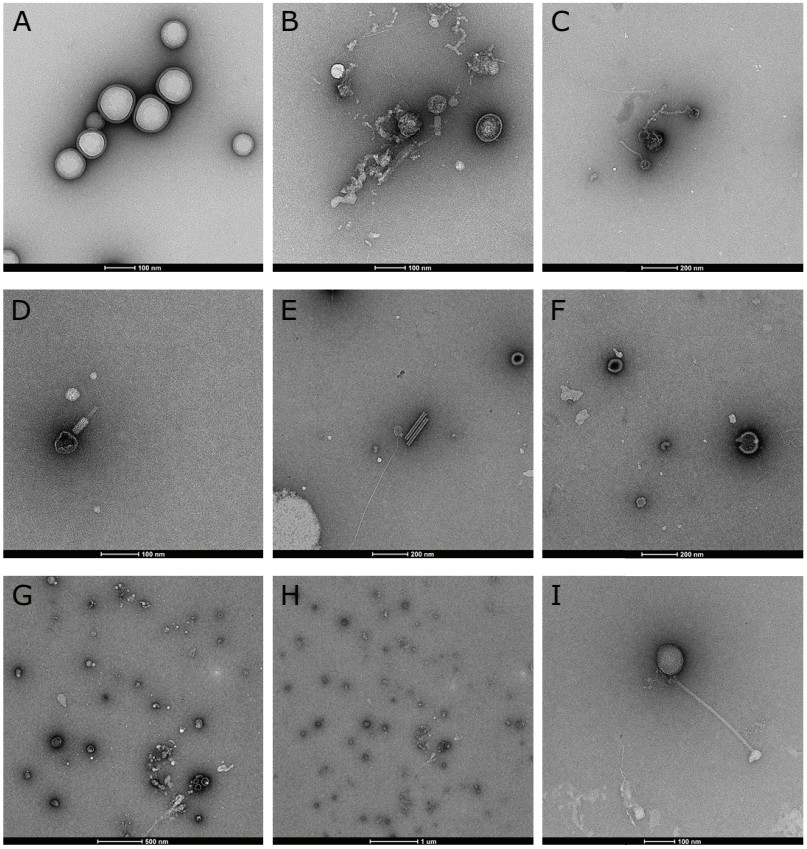

**Figure 2 Electron microscopic evidence of virus-like particles in stream biofilm samples.** (A) large morphological diversity of VLPs, including tailed bacteriophages (B, C, D, I), lemon-shaped, polyhedral (e.g. B, H), spherical and filamentous viruses was observed using TEM. Some morphologies resembled plant viruses (E) while amorphous structures (A, F, G, H) could also be membrane vesicles.

VLPs extracted by the various protocols were strongly correlated among the different samples (pairwise Spearman's $r_s$ ranging between 0.94 and 0.98).

## Transmission electron microscopy

Direct electron microscopic observations of raw biofilm samples revealed the presence and morphological diversity of virions, including tailed bacteriophages and lemon-shaped viruses, in biofilms from all three streams (Fig. 2). Polyhedral, spherical and filamentous VLPs were also observed, which may include untailed bacteriophages or viruses infecting eukaryotes. Some amorphous structures may also be interpreted as membrane vesicles.

## Virome extraction and purification

In total, we evaluated 16 protocols to concentrate, extract and purify viruses from benthic biofilms from three streams (Fig. 1). Based on the number of VLPs and DNA yields retained at the end of each protocol, we observed significant differences in relative extraction efficiencies among protocols. Across all samples, average VLP counts and DNA yields were correlated (Spearman's $r_s$ = 0.74), suggesting conformity among these two

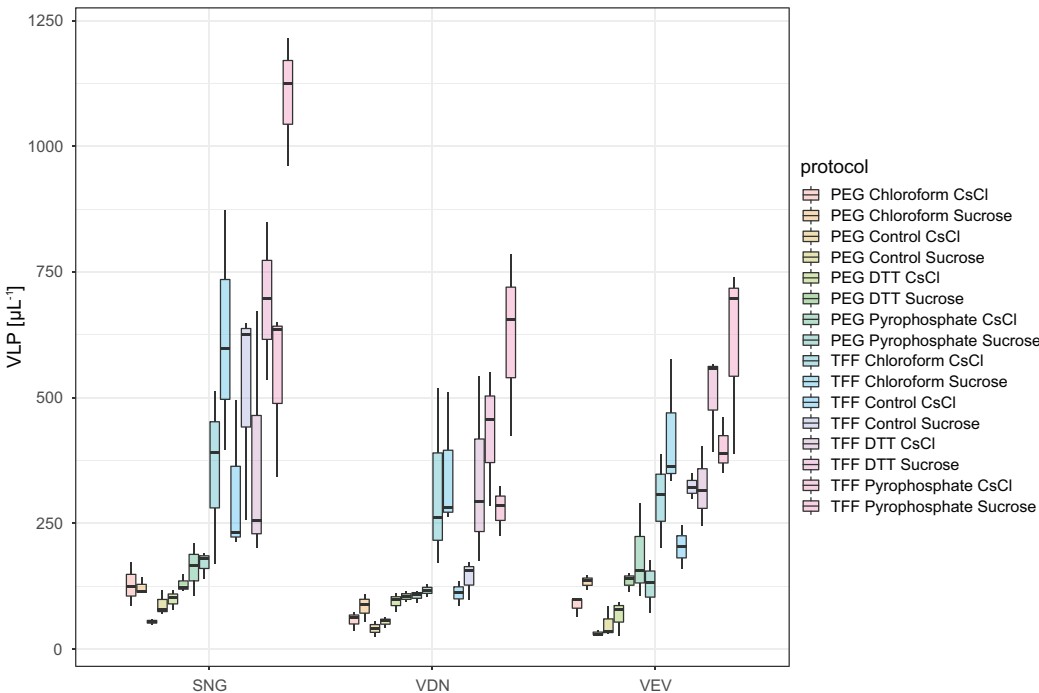

**Figure 3 VLP yields of the different combination of methods for the extraction and purification of viruses from stream biofilms.** The boxes show the number of VLPs in replicated subsamples processed with each protocol, the median is given as a horizontal line, hinges correspond to the first and third quartile while whiskers extend to the largest and smallest values counted. In each of the three streams tested, a protocol based on TFF, pyrophosphate in combination with sonication and ultracentrifugation in sucrose gradient yielded significantly higher VLP counts than any other combination of protocols.

means of evaluation. The combination of TFF for concentration, tetrasodium pyrophosphate and sonication for extraction, and sucrose gradient centrifugation for purification resulted in the highest VLP counts in all three samples (two-way ANOVA, $p < 0.01$; Fig. 3). This pipeline also yielded the highest DNA concentration (Fig. 4). Protocols involving PEG precipitation generally resulted in a lower recovery of VLPs (on average only 13.6% compared to the best performing pipeline) and DNA yields below detection limit. This may be attributable to the formation of a visible, viscous layer upon addition of PEG to the biofilm samples. The dissociation of VLPs from the biofilm matrix was most effective using tetrasodium pyrophosphate and sonication (two-way ANOVA, $p < 0.01$). Protocols based on TFF and using DTT or chloroform extracted on average 25.0% and 33.2% less VLPs than protocols using TFF followed by tetrasodium pyrophosphate treatment and sonication (Fig. 3). Samples without any physicochemical treatment to extract VLP from biofilms yielded on average 54.8% less VLPs than the tetrasodium pyrophosphate and sonication treatment. To further purify viruses, two discontinuous gradient formulations with sucrose and CsCl were tested. On average, 1.9 times more VLPs were retained by ultracentrifugation using the sucrose gradient than using the CsCl gradient in samples concentrated using TFF and treated with tetrasodium pyrophosphate and sonication (paired-$T$-Test, $p < 0.01$). This is also reflected

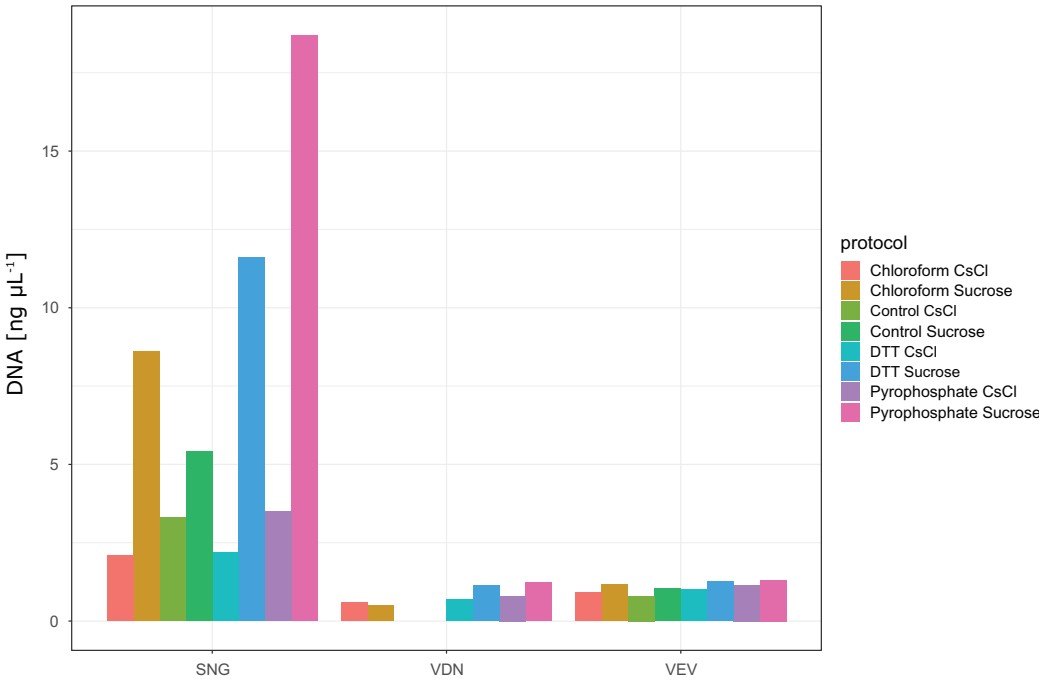

**Figure 4 DNA yields from samples processed with different protocols.** None of the protocols involving PEG precipitation resulted in detectable DNA yields (see DOI: 10.6084/m9.figshare.8943341). Protocols using TFF and either no treatment (control), chloroform, DTT, pyrophosphate in combination with sonication for extraction and either CsCl or Sucrose gradient ultracentrifugation for purification yielded detectable DNA concentrations. Highest DNA yields were obtained using a combination of TFF, followed by pyrophosphate and sonication and sucrose density gradient ultracentrifugation. Note that although VLP counts were only 1.8 times higher in the best performing protocol in SNG as compared to the other two samples (VDN, VEV), DNA yields in this sample were more than 14 times higher.

in DNA yields, which reached 1.22 ng $\mu L^{-1}$ in VDN, 1.31 ng $\mu L^{-1}$ in VEV and 18.7 ng $\mu L^{-1}$ in SNG using TFF, pyrophosphate and sonication followed by sucrose gradient centrifugation. DNA yields were on average 3 times higher using ultracentrifugation in the sucrose compared to the CsCl gradient (paired *T*-Test, *p* = 0.03) and on average 1.5, 1.7 and 1.9 times higher using tetrasodium pyrophosphate and sonication compared to DTT, no physico-chemical detachment and chloroform, respectively.

Given that stream biofilms contain abundant prokaryotic, eukaryotic and extracellular DNA, it is crucial to verify the absence of DNA potentially contaminating the samples. Negative PCR results from samples treated with DNase I confirmed the absence of DNA contamination from prokaryotic cells.

### Stream biofilm viromes

From the two sequencing runs we obtained 24388096, 22459218 and 23804190 paired-end reads from SNG, VEV and VDN, respectively (Table 2). After quality control, error correction and deduplication, on average 97.5% of the reads remained. Initial screening using taxonomer showed that human contaminant reads accounted for 0.13–0.25% of the reads, while bacterial contaminant reads accounted for 1.47–8.11% of the reads.

**Table 2  Virome dataset statistics.**

| | Raw reads | | Quality trimmed | | Contigs | Average contig length (bp) |
|---|---|---|---|---|---|---|
| | Paired-end sequences | GC content (%) | Paired-end sequences | GC content (%) | | |
| SNG | 24 388 096 | 40.36 | 24 172 084 | 40.16 | 3,698 | 452 |
| VEV | 22 459 218 | 40.51 | 22 153 818 | 40.24 | 11,323 | 645 |
| VDN | 23 804 190 | 40.68 | 23 422 408 | 40.21 | 13,591 | 676 |

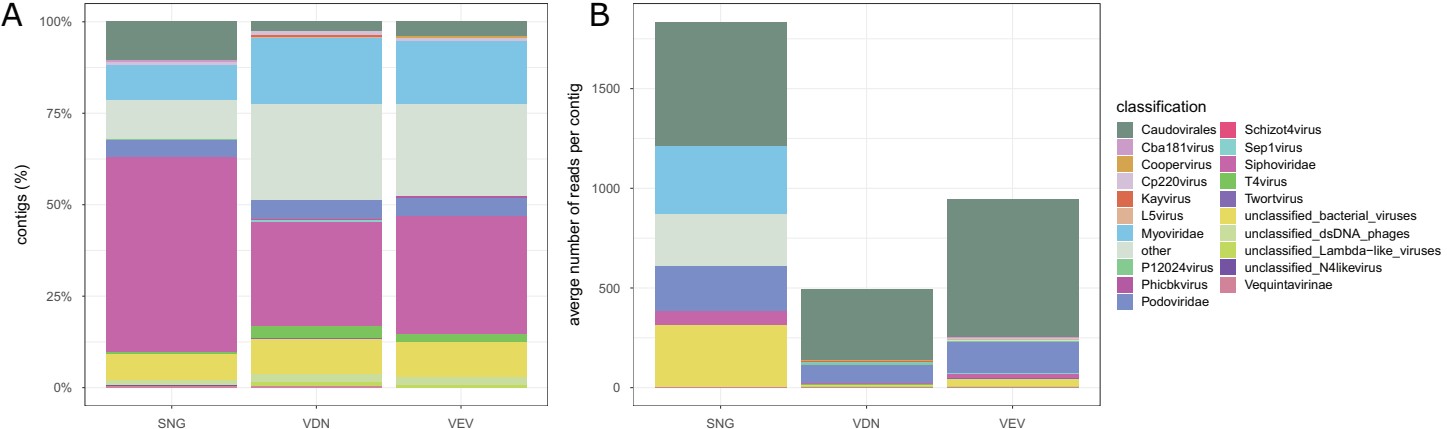

**Figure 5  Stream biofilm virome taxonomic composition.** The relative number of classified contigs among the viromes are shown (A). Stream biofilm viromes were dominated by not further classified members of *Siphoviridae* and *Myoviridae*. The relative composition of contigs was similar among the three samples, however, markedly fewer contigs were obtained from SNG than from the other two stream biofilm samples. ssDNA viruses could not be identified among contigs from any of the streams. The average number of reads mapped to classified contigs in the three samples (B).

We obtained 3,698, 11,323 and 13,591 contigs from de-novo assembly of quality-controlled and deduplicated reads from SNG, VEV and VDN, respectively. The largest contigs were 9,493, 46,665 and 54,492 bp in SNG, VEV and VDN, respectively. Hmmsearch against the PFAM databases did not yield ssDNA contigs in the three viromes. Between 726 and 2613 contigs were classified as of viral origin in the three viromes (Fig. 5). In all samples, the majority of contigs were identified as not further classified *Siphoviridae* (28.3–53.3%), followed by *Myoviridae* (9.6–18.0%) and *Podoviridae* (4.7–5.2%). Contigs classified as *T4virus*, *Cp220virus*, *Kayvirus* and *P12024virus* were common in all three biofilm viromes. However, contigs classified as *Twortvirus*, *Phicbkvirus*, *Coopervirus* and *L5virus* were only detected in viromes obtained from VEV and VDN but not in SNG. Across the three samples, on average 554.6 reads mapped to contigs classified as *Caudovirales*, averaging 63.4, 63.8 and 89.1% bp accounting for contig size in SNG, VEV and VDN, respectively. On average, 153.8 reads mapped to *Podoviridae* (range 2.2–30.8% of bp) and 115.1 reads mapped to *Myoviridae* (range 0.4–6.1% of bp), whereas only 29.4 reads mapped to *Siphoviridae* (range 1.8–17.7% of bp). There was substantial variation in the number of reads mapped to contigs in the different samples. For instance, more reads mapped on average to unclassified bacterial viruses and *Myoviridae* in SNG (311.1 and 339.7 reads accounting for 9.3 and 6.1% of bp) than in VDN

(2.6 and 2.8 reads accounting for 0.6 and 0.8% bp) or VEV (36.7 and 2.8 reads accounting for 1.5 and 0.4% bp).

## DISCUSSION

The advent of metagenomic tools has revolutionized the study of the role of viruses in numerous ecosystems (*Suttle, 2007*; *Rosario & Breitbart, 2011*; *Brum & Sullivan, 2015*; *Trubl et al., 2019*). For biofilms in general, and particularly for biofilms in streams and rivers, however, an optimized protocol for viral metagenomics has been missing. Here, we establish an optimized sample-to-sequence pipeline for the concentration and purification of viruses from stream biofilms. This protocol is publicly available at https://www.protocols.io/view/extraction-and-purification-of-viruses-from-stream-32qgqdw. We used two metrics, the number of VLP retained and the amount of DNA extracted from the samples to evaluate the performance of the different combinations of protocols (Figs. 3 and 4). This approach allowed us to obtain a relative comparison among the tested protocols. However, it does not permit the quantification of extraction efficiencies since it was not possible to enumerate VLPs without prior extraction and purification.

The best performing sample-to-sequence pipeline involves TFF for sample concentration, pyrophosphate and sonication for the detachment of viruses from the biofilm matrix and DNase I treatment followed by sucrose gradient ultracentrifugation for purification. This is similar to protocols for other complex samples, such as soils (*Trubl et al., 2016*) or marine and freshwater sediments (*Danovaro & Middelboe, 2010*). The suggested combination of protocols was consistently the best performing pipeline for biofilms obtained from three streams differing in trophic state and with different biofilm properties (Table 1). However, depending on biofilm biomass, sampling efforts should be tailored towards obtaining sufficient nucleic material for virome metagenomic sequencing. For instance, in the oligotrophic mountain stream, sampling biofilms from 0.5 m$^2$ (approximately 10–12 pebbles of 5–10 cm diameter) and using 1 L of Milli-Q water suffices to generate viromes using the optimized protocols and an appropriate library preparation strategy.

The viromes obtained using the best-performing protocol were dominated by reads of viral or unknown origin (presumably reflecting the lack of viral sequences in public databases), and contaminant sequences (i.e., of bacterial or human origin) contributed only marginally to the viromes. Following an optimized assembly strategy (*Roux et al., 2019*), the reads assembled into a large number of contigs, however, of small average contig size. Still many of the contigs were classified as of viral origin. Viral community composition was remarkably similar across the three different stream biofilms, with several contigs classified as *T4virus*, *Cp220virus*, *Kayvirus* and *P12024virus* found in all samples. Despite the use of the ACCEL-NGS® 1S PLUS DNA kit (*Roux et al., 2016*), ssDNA viruses, which account for <5% of DNA viral communities in other freshwater, marine and soil ecosystems (*Roux et al., 2016*; *Trubl et al., 2019*) could not be detected in the viromes from stream biofilms. This may be related to the choice of the density gradient harvested during the final purification step (*Kauffman et al., 2018*). Due to the low buoyant density of ssDNA viruses (*Thurber et al., 2009*), additionally sampling from, for

instance, a 1.3 g/mL CsCl density layer (*Trubl et al., 2019*) may be advisable to specifically target ssDNA viruses. Strikingly, the virome obtained from the most eutrophic stream (SNG) resulted in the lowest number of viral contigs and lacked contigs classified as *Twortvirus*, *Phicbkvirus*, *Coopervirus* and *L5virus*. However, given the low number of samples, we caution against concluding an anthropogenic effect on stream biofilm viral communities.

A range of chemical properties may explain the relative differences in virus extraction efficiency from stream biofilms. PEG precipitation generally failed to concentrate viruses from biofilm slurries, potentially due to the formation of a viscous layer that impaired PEG removal. Previously described methods for isolating viruses involve chloroform (*Marhaver, Edwards & Rohwer, 2008*; *Thurber et al., 2008*; *Hewson et al., 2012*); however, chloroform may denature the lipid envelopes surrounding viral capsids, internal lipid membranes. Nucleo-cytoplasmic large DNA viruses (NCLDV), including Phycodnaviridae, which predominantly infect freshwater and marine algae, may thus be sensitive to chloroform treatment (*Feldman & Wang, 1961*). Similarly, DTT is a reducing agent, which breaks disulfide bonds in proteins. This may explain the reduced recovery of VLPs from samples treated with PEG, chloroform or DTT.

Extracellular DNA is a common component of the biofilm matrix (*Flemming & Wingender, 2010*), which together with DNA from damaged eukaryotic and prokaryotic cells needs to be removed prior to virome sequencing. We propose a DNase I treatment as an efficient way to digest extracellular DNA from biofilms, followed by sucrose density gradient ultracentrifugation for further purification. However, DNase I will not degrade ssDNA or RNA and recent studies used nuclease cocktails including DNases, RNases and Benzoases (*Temmam et al., 2015*; *Rosario et al., 2018*) to eliminate various contaminant nucleic acids. The treatment order is important because otherwise DNA from viral particles damaged during ultracentrifugation may be digested by the DNase digestion. Compared to ultracentrifugation in a CsCl gradient, sucrose gradient ultracentrifugation probably maximized the purification of a wider range of viruses because of the larger gradient of densities recovered with this method. Moreover, CsCl density gradient separation may exclude viral particles as that may be too buoyant (*Thurber et al., 2009*; *Kauffman et al., 2018*), or degrade the structure of enveloped viruses (*Lawrence & Steward, 2010*) and therefore reduce their recovery. Critical steps of virome preparation concerns nucleic acid extraction and library preparation. We chose a derivation of a standard extraction protocol, which allows the extraction of both ssDNA and dsDNA viruses (*Sambrook, Fritsch & Maniatis, 1989*). Optimization of DNA extraction protocols, such as done for soil viromes (*Trubl et al., 2016*), may be necessary for biofilms sampled from environments with high humic acid concentrations such as in boreal streams or streams draining wetlands, potentially causing inhibition during library preparation.

Finally, there is a plethora of tools available for the bioinformatic analysis of viromes (e.g., those implemented in iVirus (*Bolduc et al., 2017*) and available through powerful computational infrastructures such as CyVerse (www.cyverse.org) or KBase (www.kbase.us). Here, we opted for contig assembly, classification and read mapping to assess the capability of the laboratory procedures to generate diverse viromes from stream

biofilm. Clearly, the choice of bioinformatic analyses depends on the specific questions regarding the composition and role of viruses in stream biofilms.

## CONCLUSIONS

In conclusion, we provide a first protocol for the generation of viromes from stream biofilms. The sample-to-sequence pipeline generates diverse viromes, however the purification scheme may select against viruses with different buoyant densities, such as ssDNA viruses or viruses containing lipids. Similarly, filtration may discriminate against large or filamentous viruses and some viruses may be sensitive to chemicals used during biofilm breakup. However, this is a first step towards a better understanding of the roles viruses may play in stream ecology. By providing a step-by-step protocol on protocols.io, we hope to further stimulate research on phage diversity in stream biofilms.

## ACKNOWLEDGEMENTS

We acknowledge the help of Davide Demurtas for TEM imaging at EPFL (CIME).

### Funding

This work was supported by the Swiss National Science Foundation (No. 169651) to Tom J. Battin and Hannes Peter. The funders had no role in study design, data collection and analysis, decision to publish, or preparation of the manuscript.

### Grant Disclosures

The following grant information was disclosed by the authors:
Swiss National Science Foundation: 169651.

### Competing Interests

The authors declare that they have no competing interests.

### Author Contributions

- Meriem Bekliz conceived and designed the experiments, performed the experiments, analyzed the data, contributed reagents/materials/analysis tools, prepared figures and/or tables, authored or reviewed drafts of the paper, approved the final draft.
- Jade Brandani performed the experiments, contributed reagents/materials/analysis tools, authored or reviewed drafts of the paper, approved the final draft.
- Massimo Bourquin analyzed the data, authored or reviewed drafts of the paper, approved the final draft.
- Tom J. Battin conceived and designed the experiments, authored or reviewed drafts of the paper, approved the final draft.
- Hannes Peter conceived and designed the experiments, performed the experiments, analyzed the data, contributed reagents/materials/analysis tools, prepared figures and/or tables, authored or reviewed drafts of the paper, approved the final draft.

## DNA Deposition

The following information was supplied regarding the deposition of DNA sequences:

The raw sequences are available at the European Nucleotide Archive: ERP116349.

## Data Availability

Peter, Hannes; Bekliz, Meriem (2019): Stream biofilm viromes. figshare. Dataset. DOI 10.6084/m9.figshare.8943341.v2.

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
