# Peer review of "Benchmarking protocols for the metagenomic analysis of stream biofilm viromes"

_PeerJ, doi:10.7717/peerj.8187_

## Round 0.1 · original submission · Major Revisions

Please address all the concerns raised by the reviewers - see also the detailed comments of reviewer 1 (annotated manuscript), including the criticisms that the employed CsCl density gradients will miss a large portion of viruses.

·

Basic reporting

Overall the writing is good, but more work needs to go into explaining the methods. I have included notes in my review. Environmental virus literature is well cited and review, but in the introduction there is to much focus on bulk metagenomes and not enough on virus targeted metagenomics (viromics). All the raw data is available. I made suggestions for the figures in my review and I also recommend clustering the viruses into viral populations.

Experimental design

The overall design is correct and the optimization efforts are a good addition. The authors did a lot of work to target both ssDNA and dsDNA viruses including reading and citing relevant literature, and using the correct library kit and amplification methods, but did not target the correct CsCl density range to capture ssDNA and much of the dsDNA viruses. The targeted bioinformatic analyses for ssDNA are not well described. Please go through the methods section and be more descriptive of all the approaches.

Validity of the findings

The work here is novel and interesting and well help future viral analyses of freshwater bio films. Some of the discussion and conclusions need to be re-evaluated based on my suggestions.

Additional comments

This paper has a lot of solid work and the figures are appealing. It is a lot for any reader to digest, so please work on making it a little clearer.

·

Basic reporting

This paper describes the development and validation of a methodological pipeline for extracting viruses from stream biofilms for the purposes of virus enumeration and virome sequencing. While none of these techniques are novel, the authors are clear that they are adapting previously described protocols for a unique application (specifically, stream biofilms). All data are accessible and the final protocol is made available on protocols.io

Experimental design

Given that this is a methods paper, there were a few areas where I felt details were lacking and/or references to prior protocols are needed. I've detailed them below:
1) Line 88 - what volume was used to collect the biofilm and processed downstream? Is there a standard volume suggestion relative to surface area of stones?
2) Line 95 - is there a reference that has previously used the tetrasodium pyrophosphate and sonication method?
3) Line 121 - verify that 100 xg is correct? Seems very slow...
4) Line 153 - is there a reference for treatment with DTT and why this concentration was chosen?
5) Line 166 – provide a reference for the 16S rRNA gene primers. Also, were any tests performed to determine if there are inhibitors present in the DNA produced when following this protocol?
6) Lines 177 and 180 – how much volume was harvested as the viral fraction from the CsCl or sucrose ultracentrifugation?
7) Line 179 – provide a reference for the sucrose densities used? Protocols.io includes a link to a sucrose cushion protocol but it’s for pelleting viruses through a sucrose cushion, not layering and capturing the viruses between the layers as is done in this protocol.
8) Line 188 – add the concentration of SYBR Gold used
9) Line 200 – “purified viral samples were suspended in” is a little misleading because it makes it sound like the viral particles are pelleted or not contained in a volume of liquid. Possibly better to just say that 0.1 volume of TE, etc… was added to the purified samples
10) Line 202 – is this cold EtOH that is added?
11) Reference for the CTAB protocol?


Minor text edits:
1) Line 88, "benthic biofilm" should be "benthic biofilms"
2) Line 97, reword "worked fine for us previously" to "has been previously established"
3) Throughout the manuscript, apostrophes are often used instead of commas in numbers (e.g., line 111 but this happens throughout)
4) Line 116: Reword to say "Extraction aims to liberate VLP from the biofilm matrix, while purification aims to reduce the amount of contaminating nucleic acids and cellular debris.
5) Line 278 – “This pipeline also yielded the highest DNA concentration” – add “the”
6) Lines 258-364: These sentences are confusing, please reword for clarity. Why does it matter that ssDNA viruses lack a lipid envelope for their susceptibility to chloroform? Add a reference for the fact that some tailed phage are sensitive to chloroform.
7) Line 361: Phycodnaviridae should be italicized
8) Line 379: I found the ending of this section to be pretty abrupt – maybe add a few sentences about the library construction, sequencing, bioinformatics?
9) The reference formatting is very inconsistent – some journal titles are abbreviated, others are written out fully. Some titles are all first letter capitals, others are sentence-style capitalization.
10) Figure 2 caption – I don’t see some of the morphologies mentioned in the caption, such as the lemon-shaped particles. I suggest labeling each panel of this figure and referring to each one by letter after the corresponding morphology in the legend. In addition, the center panel appears to be similar to a plant virus in morphology, which might be worth mentioning in the list of morphologies.
11) Figure 5: It is very difficult to distinguish the different colors in this figure. Also, I believe this graph refers to the number of contigs belonging to each viral type, but it would also be useful to see the number of reads mapping to contigs belonging to each viral type because otherwise you could have 1 highly represented full genome and that will get a much smaller piece of the graph than something with many contigs but low coverage.

Validity of the findings

I have three concerns with this paper:
1) The use of only DNase I to eliminate contaminating nucleic acids is fine for viromes focused on dsDNA; however, it should be noted (at least in the discussion) that DNase I will not degrade ssDNA or RNA. Most recent studies utilize a nuclease cocktail (e.g., Temmam et al. "Host-associated metagenomics: a guide to generating infectious RNA viromes." PLoS One 10, no. 10 (2015): e0139810 or Rosario et al. "Diversity of DNA and RNA viruses in indoor air as assessed via metagenomic sequencing." Environmental science & technology 52, no. 3 (2018): 1014-1027." This does not seem to have affected the authors conclusions, since they did not see a lot of ssDNA viruses and RNA was not targeted, but is important to mention since others will likely follow this protocol.

2) The TEM images of the viruses shown were obtained from unprocessed biofilm samples (as described in lines 106 - 111). First, I'm not sure that all of the images are actually of viruses - many seem amorphous, like they could be membrane vesicles. While I agree that it's impossible to get definitive identification, the possibility that they could be vesicles should be addressed. Second, (and if there is ANY possible way the authors could generate these images, I would strongly encourage it), TEM should be performed on the final purified VLPs. This is important because it would show if some of "morphologies" seen in the initial TEM from the unprocessed sample disappear throughout the processing (like vesicles would) and second, because it will reveal if any viral types are lost throughout the processing. If it is not possible to obtain TEM images from the final processed samples, I recommend removing the images altogether, since it's not very relevant if you don't show how they are changed/maintained through the purification protocol.

3) The potential biases of the purification scheme need to be discussed. For example, Kauffman et al. (2018; A major lineage of non-tailed dsDNA viruses as unrecognized killers of marine bacteria. Nature, 554(7690), p.118) showed that the use of density gradients biased against some common groups of non-tailed viruses. I imagine it is possible that the sucrose gradient used in this paper might bias against certain groups of viruses (for example, viruses with lipids or ssDNA viruses with different densities?). While it is impossible to have a method for viral purification that isn’t biased in some manner, and it’s not practical to sequence from every virome preparation method, it is critical to discuss the potential limitations of the optimized protocol.

---

## Round 0.2 · Minor Revisions

Please incorporate the latest minor suggestions made by the reviewers.

·

Basic reporting

I am very pleased with the efforts of the authors to address the reviewers' comments. The manuscript is much clear and reads well. I think the manuscript is ready for publication, but did provide some additional comments.

1. Per ICTV guidelines virus names should never be italicized. https://talk.ictvonline.org/information/w/faq/386/how-to-write-virus-and-species-names

2. Lines 286–288 “We mapped the reads to the contigs using BWA-MEM (http://bio-bwa.sourceforge.net/) and then counted the number of reads mapping each contig using samtools (Li et al. 2009).”

What was the mapping threshold and was is the default parameters for BWA-MEM?

3. Lines 359–362 “Across the three samples, on average 554.6 reads mapped to contigs classified as Caudovirales, 153.8 mapped to Podoviridae and 115.1 reads mapped to Myoviridae, whereas only 29.4 reads mapped to Siphoviridae. There was substantial variation in the number of reads mapped to contigs in the different samples.”

I think it would be better to present the average % and no the actual number of the reads that mapped.

4. Lines 441–442 “Finally, there is a plethora of tools available for the bioinformatic analysis of viromes (e.g. those implemented in iVirus (Bolduc et al., 2017).”

I know Ben would appreciate the shout out because he has worked very hard on making tools available for everyone. Really though, his efforts are because of the various grants Dr. Bonnie Hurwitz was awarded and DOE funding. To be more transparent and make other researchers have to work less you should direct them to CyVerse (https://www.cyverse.org/) and KBase (https://kbase.us/). iVirus was started building off iMicrobe, which built off iPlant (the original app). Because so many tools are used in common, CyVerse brought them all together. KBase is just like CyVerse, but has some different tools, is supposed to be more user friendly, and because it is linked to government funding it *should* be sustained indefinitely. I think both are good options that can aid researchers. Maybe keep it as is and direct people to CyVerse and KBase.

Experimental design

Author's address my comments and it reads well.

Validity of the findings

This paper will be a nice addition to current literature and is robust.

·

Basic reporting

The authors have addressed my major concerns. I have made a few edits on the manuscript itself, there were several mistakes in the newly added text.

Experimental design

No comment

Validity of the findings

No comment

---

## Round 0.3 · accepted · Accept

Thank you for incorporating the reviewers suggestions and for updating your manuscript. Congratulations on the acceptance of your paper.